# Detection of SARS-CoV-2 RNA by direct RT-qPCR on nasopharyngeal specimens without extraction of viral RNA

**Mohammad Rubayet Hasan**[1,2]*, **Faheem Mirza**[1], **Hamad Al-Hail**[1],
**Sathyavathi Sundararaju**[1], **Thabisile Xaba**[1], **Muhammad Iqbal**[1], **Hashim Alhussain**[3], **Hadi Mohamad Yassine**[3], **Andres Perez-Lopez**[1,2], **Patrick Tang**[1,2]

**1** Department of Pathology, Sidra Medicine, Doha, Qatar, **2** Weill Cornell Medical College-Qatar, Doha, Qatar, **3** Qatar University, Doha, Qatar

* mhasan@sidra.org

**Data Availability Statement:** All relevant data are within the manuscript and its Supporting Information files.

**Funding:** The authors received no specific funding for this work.

## Abstract

To circumvent the limited availability of RNA extraction reagents, we aimed to develop a protocol for direct RT-qPCR to detect SARS-CoV-2 in nasopharyngeal swabs without RNA extraction. Nasopharyngeal specimens positive for SARS-CoV-2 and other coronaviruses collected in universal viral transport (UVT) medium were pre-processed by several commercial and laboratory-developed methods and tested by RT-qPCR assays without RNA extraction using different RT-qPCR master mixes. The results were compared to that of standard approach that involves RNA extraction. Incubation of specimens at 65˚C for 10 minutes along with the use of TaqPath™ 1-Step RT-qPCR Master Mix provides higher analytical sensitivity for detection of SARS-CoV-2 RNA than many other conditions tested. The optimized direct RT-qPCR approach demonstrated a limit of detection of $6.6 \times 10^3$ copy/ml and high reproducibility (co-efficient of variation = 1.2%). In 132 nasopharyngeal specimens submitted for SARS-CoV-2 testing, the sensitivity, specificity and accuracy of our optimized approach were 95%, 99% and 98.5%, respectively, with reference to the standard approach. Also, the RT-qPCR $C_T$ values obtained by the two methods were positively correlated (Pearson correlation coefficient r = 0.6971, $p$ = 0.0013). The rate of PCR inhibition by the direct approach was 8% compared to 9% by the standard approach. Our simple approach to detect SARS-CoV-2 RNA by direct RT-qPCR may help laboratories continue testing for the virus despite reagent shortages or expand their testing capacity in resource limited settings.

## Introduction

The ongoing pandemic of coronavirus disease (COVID-19) caused by a novel coronavirus, severe acute respiratory syndrome coronavirus 2 (SARS-CoV-2) posed an unprecedented public health threat to the entire world. In the absence of an effective vaccine or specific treatment against the virus, early detection and contact tracing, physical distancing measures and

**Competing interests:** The authors have declared that no competing interests exist.

quarantine of cases are considered the cornerstones to curb the community transmission of SARS-CoV-2 [1–3]. Since the virus was identified and its genome sequenced in early January 2020, detection of viral RNA in respiratory specimens by real-time reverse transcription PCR (RT-qPCR) remains the main approach to manage the outbreak by allowing early detection of cases and targeted measures to prevent transmission of the virus [4]. The massive demand for SARS-CoV-2 RT-qPCR has brought about a global shortage and supply chain irregularities of RNA extraction kits that are crucial for RT-qPCR testing [5–7]. Detection of viral pathogens directly from clinical specimens without RNA extraction has been described earlier in cases such as detection of norovirus from fecal specimens [8], human papilloma virus from crude cell extracts [9], and Zika virus from blood or serum samples [10]. In this study, we tested a number of commercial and laboratory-developed, specimen pre-treatment procedures to optimize the performance of direct RT-qPCR for SARS-CoV-2 avoiding the RNA extraction step. This method was validated against a standard approach that included extraction of viral RNA on a commercial automated extraction platform.

## Materials and methods

Standard approach for detection of SARS-CoV-2 RNA from nasopharyngeal specimens in our laboratory involves extraction of total nucleic acids from specimens in an IVD-labeled, automated extraction platform followed by RT-qPCR, based on one of the assays (Table 1) suggested by World Health Organization (WHO) [11]. The performance standards of the assay were established in our laboratory according to College of American Pathologists (CAP) standards, and at time of writing this paper, the assay was used to test more than 2000 respiratory specimens to screen patients for potential infection with SARS-CoV-2. Therefore, the assay was used as the reference method for all other alternative approaches assessed in this study. All specimen preparations, pre-treatments and PCR setup were performed in a Class II Biosafety cabinet in a Biosafety level 2 (BSL2) facility.

Clinical specimens: Nasopharyngeal flocked swab (NPFS) specimens collected in universal viral transport (UVT) medium (Becton, Dickinson and Company) (n = 180) submitted for testing SARS-CoV-2 or other respiratory viruses at Sidra Medicine, Doha, Qatar were used in this study. To maintain patient anonymity, each sample was coded, and all patient identifiers were removed to ensure that personnel involved in this study were unaware of any patient information. Ethics approval was not sought because the study involves laboratory validation of test methods and the secondary use of anonymous pathological specimens that falls under the category 'exempted' by Sidra Medicine Institutional Review Board. For spiking experiments, NPFS specimens (n = 6) were collected from laboratory volunteers after obtaining

**Table 1. Direct RT-qPCR on SARS-CoV-2 positive and negative NPFS specimens after heating at 65°C for 10 minutes with different RT-qPCR master mixes.**

| Sample No. | SARS-CoV-2 $C_T$ | | | |
|---|---|---|---|---|
| | **Standard method** | **Quantifast Pathogen RT-PCR + IC Master Mix** | **PrimeDirect™ Probe RT-qPCR Mix** | **TaqPath™ 1-Step RT-qPCR Master Mix** |
| 1 | 21.5 | 29.4 | 24.6 | 22.8 |
| 2 | 34.5 | Undetermined | Undetermined | 35.3 |
| 3 | 24.5 | 30.4 | 28.7 | 25.5 |
| 4 | 22 | 29.9 | 31.4 | 25.8 |
| 5 | Undetermined | Undetermined | Undetermined | Undetermined |

NPFS specimens were either subjected to viral RNA extraction by standard method using the NucliSENS easyMAG automated extraction system (bioMerieux), or diluted 4-fold with NFW followed by incubation at 65°C for 5 minutes. All samples were tested for SARS-CoV-2 RNA by standard RT-qPCR using different master mixes in duplicate and mean $C_T$ values were compared.

written informed consents. No personal data were collected, and specimens were labeled with random numbers so that test results cannot be linked to an individual.

Nucleic acid extraction: Nucleic acids from 0.2 ml of NPFS specimens were extracted on a NucliSENS® easyMAG platform (bioMérieux, France) according to the methods described by the manufacturers.

Pre-treatment of specimens: Unless otherwise stated, specimens were diluted using nuclease free water. Heat treatment of specimens at different temperature was performed in a ThermoMixer (Eppendorf). Specimens processed with Arcis Coronavirus RNA extraction research kit (Arcis Biotechnology) were performed according to manufacturer's instructions. Briefly, 90 μl specimen was mixed with 6 μl Reagent 1 RTU and either left unheated or heat lysed at 60°C for 5 minutes. 40 μl of lysate was then mixed with 2 μl of Reagent 2a and was used directly for Rt-qPCR. For Takara PrimeDirect™ Probe RT-qPCR Mix (Takara Bio), 0.2 ml specimens were heated at 99°C for 10 mins and then centrifuged at 4000rpm for 5 min. The supernatants were collected and 7μl of supernatant was directly used in RT-qPCR reaction mixture.

RT-qPCR: Primers and probes for detection of SARS-CoV-2, HCoV-HKU1, RNaseP and MS2 bacteriophage are listed in S1 Table [4,12–14]. For detection of HCoV-HKU1 using Quantifast Pathogen RT-PCR + IC kit (Qiagen), five μl of extracted or pre-treated samples were mixed with 20 μl of a master mix containing 5 μl of Quantifast Pathogen RT-PCR + IC Master Mix, 0.25 μl of Quantifast RT Mix and 0.5 μl of 50 x ROX dye solution and primers and probes to final concentrations shown in S1 Table. Thermal cycling was performed in a ABI7500 Fast instrument (Thermofisher Scientific) with 1 cycle of reverse transcription at 50°C for 20 min followed by 1 cycle of PCR activation at 95°C for 5 min, followed by 40 amplification cycles each consisting of 95°C-15s and 60°C-60s. For detection of HCoV-HKU1 using TaqPath™ 1-Step RT-qPCR kit (Thermofisher Scientific), five μl of extracted or pre-treated samples were mixed with 5 μl of a master mix containing 5 μl of TaqPath™ 1-Step RT-qPCR Master Mix and primers and probes to final concentrations shown in S1 Table. Thermal cycling was performed in a ABI7500 Fast instrument (Thermofisher Scientific) with 1 cycle of reverse transcription at 50°C for 15 min followed by 1 cycle of polymerase activation at 95°C for 2 min, followed by 40 amplification cycles each consisting of 95°C-15s and 60°C-60s.

For detection of SARS-CoV-2 with standard method, five μl of nucleic acid extracts from NucliSENS® easyMAG system were mixed with 7.5 μl of a master mix containing 2.5 μl of Quantifast Pathogen RT-PCR + IC Master Mix, 0.125 μl of Quantifast RT Mix and 0.25 μl of 50 X ROX dye solution and primers and probes to final concentrations shown in S1 Table. RNA extraction and PCR inhibition were monitored by an internal control PCR assay or an RNaseP assay, using the primers and probes shown in S1 Table. Specimens were spiked with a tittered preparation of MS2 bacteriophage to serve as a template for internal control assay. For direct PCR on specimens, pre-treated specimens were assessed in the same way except that extracted MS2 bacteriophage RNA was mixed with PCR master mix to serve as an inhibition control. A synthetic RNA (IDT) based on E-Sarbeco assay [4] amplicon sequence was used as a positive control. As negative control, 0.2 ml neonatal calf serum (NCS) (Thermofisher Scientific) was extracted along with specimens and used with each PCR run. SARS-CoV-2 RT-qPCR with Takara PrimeDirect™ Probe RT-qPCR Mix was performed as described by the manufacturer. Briefly, 7 μl of processed samples were mixed with 18 μl of master mix containing 12.5 μl of PrimeDirect Probe RT-qPCR Mix and primers and probes to final concentrations shown in Table 1. Thermal cycling was performed in a ABI7500 Fast instrument (Thermofisher Scientific) with 1 cycle of denaturation at 96°C for 10 sec, 1 cycle of reverse transcription at 60°C for 5 min followed by 45 amplification cycles each consisting of 95°C-5s and 60°C-30s. SARS-CoV-2 RT-qPCR with TaqPath™ 1-Step RT-qPCR kit with the optimized

approach (specimens heat treated at 65°C for 10 min) was performed as above except that 8 μl of specimen was used per 20 μl reaction.

A total of 132 NPS that were previously tested by the standard approach were also tested by the optimized, direct approach and the results were compared. An additional 30 NPS was tested in the same way except that RNaseP was detected as an internal control or specimen control instead of MS2 bacteriophage. In addition, an external quality assessment (EQA) panel (8 specimens) from Quality Control for Molecular Diagnostics (QCMD) was tested simultaneously with the standard and the direct approach, as well as with a QIAstat-Dx Respiratory 2019-nCoV Panel (Qiagen) and the results were compared with expected results.

Limit of detection (LOD) study: A SARS-CoV-2 positive nasopharyngeal specimen ($C_T$ = 23.3) was titered by the optimized approach using synthetic RNA, and serially diluted (up to $5 \times 10^{-7}$—fold) using a negative specimen. All diluted specimens were pre-treated by heating at 65°C for 10 min and then assessed by SARS-CoV-2 RT-qPCR in replicates of 8. The $C_T$ values obtained were used to calculate limit of detection (LOD) and intra-assay reproducibility of direct RT-qPCR.

Statistical analysis: Sensitivity, defined as the number of true positive results divided by the sum of true positive and false negative results; specificity, defined as the number of true negative results divided by the sum of true negative and false positive results; and accuracy (concordance), defined as the sum of true positive and true negative results divided by the total number of test samples, were calculated and expressed as percentages. Ninety-five percent confidence intervals (CI) for sensitivity, specificity and accuracy were calculated by the Clopper-Pearson interval or exact method using an online, diagnostic test evaluation calculator (MedCalc, 2018). Correlation between the RT-qPCR CT values between standard approach and optimized direct approach was determined by Pearson's coefficient calculator. The limit of detection with 95% endpoint ($C_{95}$) was determined by Probit regression analysis [15].

## Results

Using a human coronavirus HKU1 (hCoV-HKU1) positive specimen as a surrogate for SARS-CoV-2, we first assessed whether specimens can be used directly for RT-qPCR after 2–10 fold dilution with nuclease free water (NFW), simple heat treatment (100°C for 5 min) and centrifugation to remove any insoluble material that may be present in the specimen. The pre-treated specimens were then assessed in parallel with extracted specimens by a previously described, laboratory developed RT-qPCR for HCoV-HKU1 (S1 Table). A significant loss of sensitivity was observed with a RT-qPCR $\Delta C_T$ ranging from 10–14 (S2 Table). To determine whether any components of UVT medium (Becton, Dickinson and Company) were inhibitory to RT-qPCR, we collected nasopharyngeal flocked swabs (NPFS) from laboratory volunteers in NFW along with swabs in UTM. We then spiked a SARS-CoV-2 positive specimen to all specimens and assessed them by SARS-CoV-2 RT-qPCR. However, no significant improvement in sensitivity was observed (S3 Table).

Similar results were observed when two commercial test kits were used for direct RT-qPCR: Arcis Coronavirus RNA extraction research kit comes with lysis reagents that can be used directly in RT-qPCR; and Takara PrimeDirect™ Probe RT-qPCR kit provides a master mix that is compatible with heat-treated specimen extracts. However, in our evaluation both test kits failed to demonstrate an acceptable level of sensitivity (S4 and S5 Tables). Our attempts to further optimize the pre-treatment conditions showed modest improvement with a non-ionic detergent, Tween-20, and further improvement with a heating step at 65°C for 10 min without centrifugation ($\Delta C_T$ = 5.2) (S6 Table). We then tested this low heat approach with more SARS-CoV-2 positive specimens and using 3 different RT-qPCR master mixes.

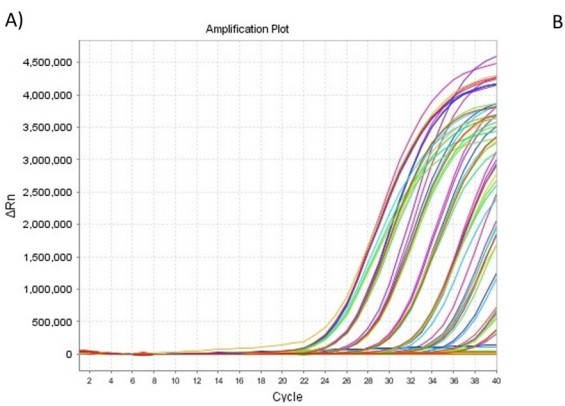
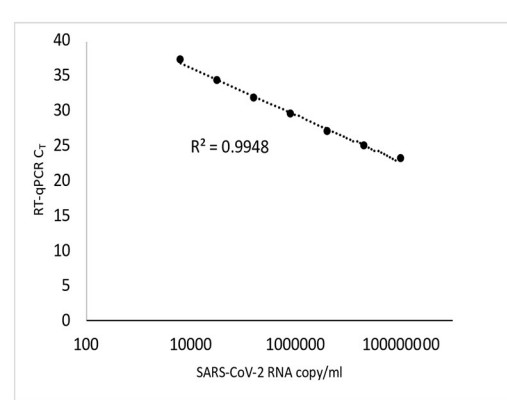

**Fig 1. Linearity of SARS-CoV-2 RT-qPCR by direct approach.** A tittered SARS-CoV-2 positive nasopharyngeal specimen was serially diluted using a negative specimen, pre-heated at 65˚C for 10 min and assessed by SARS-CoV-2 RT-qPCR as described in the Supplemental methods. A) Amplification curves from SARS-CoV-2 RT-qPCR assay on the serially diluted sample. B) RT-qPCR $C_T$ values were plotted against estimated copy number of SARS-CoV-2 RNA in each dilution. Each data point represents an average of data obtained from 8 replicates.

Interestingly, we found that with TaqPath™ 1-Step RT-qPCR Master Mix, 4/4 positive samples were correctly detected with a $\Delta C_T$ range 0.8–3.8 (Table 1). On the other hand, two other master mixes gave higher $\Delta C_T$ and 1/4 positive results were missed by both. Based on these results, the optimal pre-treatment and reaction conditions for the direct approach were: i) transfer and dilute (4-fold) 10 µl of NPFS specimen in NFW; ii) incubate at 65˚C for 10 min; and iii) test 8 µl of heat lysed specimen in a 20 µl reaction using TaqPath™ 1-Step RT-qPCR Master Mix.

The analytical sensitivity of the direct RT-qPCR assay using specimens prepared in this manner was determined by serially diluting a specimen positive for SARS-CoV-2 with a negative specimen as a diluent. The positive specimen was titered based on the $C_T$ value obtained by the direct approach using a standard curve prepared with SARS-CoV-2 synthetic RNA. The $C_T$ values were linear ($R^2$ = >0.99) over the range of $10^3$ copy/ml to $10^8$ copy/ml, the highest concentration used in this analysis (Fig 1A and 1B). The 95% limit of detection (LOD; $C_{95}$) of the assay with the direct approach was $6.6 \times 10^3$ copy/ml. The $C_T$ variation between RT-qPCR replicates across different concentration of analytes were <1 with an average coefficient of variation (CV%) of 1.2%.

A total of 132 NPFS specimens that were previously tested with standard approach including viral RNA extraction were re-tested with the new direct approach. The direct approach detected all except one positive case with $C_T > 38$. On the other hand, the direct approach detected ($C_T > 37$) SARS-CoV-2 in one specimen that was negative by standard approach. Overall agreement of results between two approaches was >98% (Kappa = 0.939; 95% CI = 0.854 to 1.000). The sensitivity and specificity of the new approach compared to the reference method were 95% and 99%, respectively (Table 2). The RT-qPCR $C_T$ values for all specimens that were positive by both methods (n = 18) were positively correlated with a Pearson coefficient (R) of 0.6971 ($p < 0.01$) (Fig 2). The rate of PCR inhibition among the specimens that gave negative RT-qPCR results by the direct approach was 8% compared to 9% by the standard approach. The direct approach accurately detected SARS-CoV-2 RNA in all except one specimen in an external quality assessment (EQA) panel provided by Quality Control for Molecular Diagnostics (QCMD) (S7 Table). The specimen that gave discrepant result was reported as 'borderline' by QCMD, and SARS-CoV-2 RNA was also undetectable in this

**Table 2. Performance of optimized direct RT-qPCR approach with reference to standard approach for detection of SARS-CoV-2 RNA.**

| Statistic | Value | 95% CI |
|---|---|---|
| Total number of specimens | 132 | - |
| True positive | 18 | - |
| True negative | 112 | - |
| False positive | 1 | - |
| False negative | 1 | - |
| Sensitivity | 95.0% | 74% to 99.8% |
| Specificity | 99.0% | 95.2% to 99.9% |
| Accuracy | 98.5% | 94.6% to 99.8% |

specimen by standard approach and by a commercial test, QIAstat-Dx Respiratory 2019-nCoV Panel (Qiagen).

Finally, we also tested our direct RT-qPCR approach using RNaseP as an internal control or a specimen quality control within a duplex RT-qPCR with SARS-CoV-2 and compared the results with that of the standard method involving RNA extraction (S8 Table). In 30 NPFS specimens, RNaseP was detected in all specimens, but with an average loss of $C_T$ ($\Delta C_T$) by 4.2 ±1.7 as compared to an average loss of SARS-CoV-2 $C_T$ by 2±1.3. Surprisingly, RNaseP was not detected in two samples by the standard method that were positive for RNaseP by the direct method. This could potentially be due to interfering substances concentrated by the

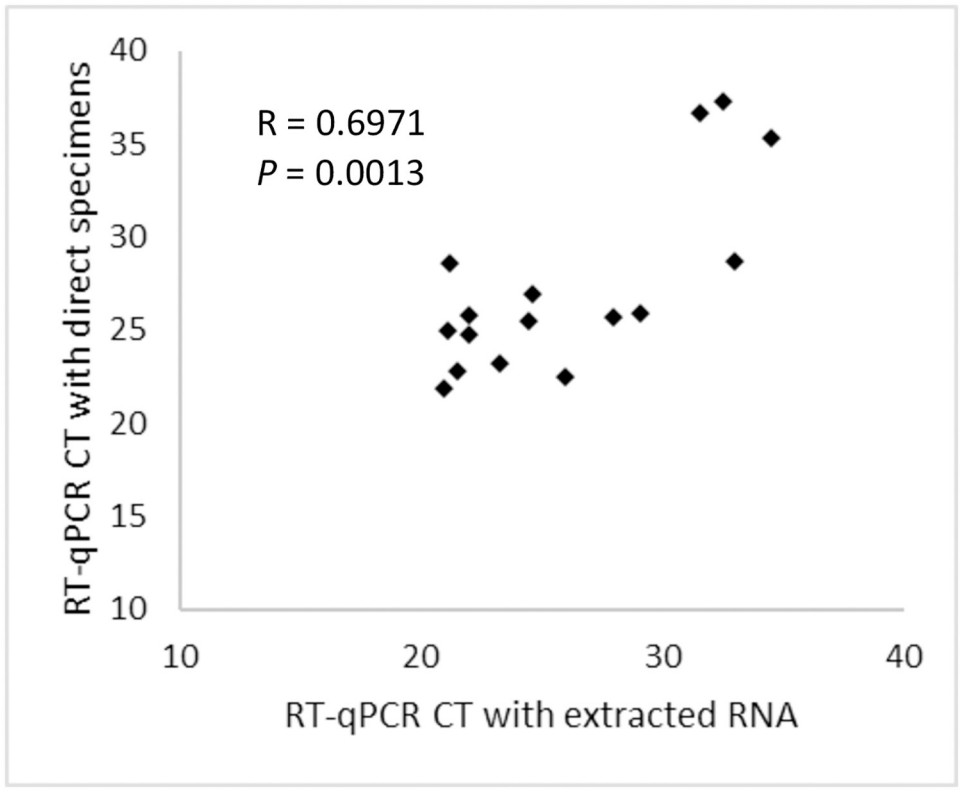

**Fig 2. Correlation of RT = qPCR $C_T$ values obtained by direct versus standard approach.**

extraction process. In this set, one sample that was positive for SARS-CoV-2 by standard method with high $C_T$ (36.3) was missed by the direct method.

## Discussion

Success in RT-qPCR testing depends on multiple factors. RNA extraction is preferable to the use of direct specimens because the extraction process concentrates and purifies the RNA targets and excludes PCR inhibitory substances. The use of pre-treated or untreated specimens directly in RT-qPCR is challenging because of the presence of inhibitors and RNA loss due to heating and/or RNases. After many attempts with various pre-treatment agents and conditions, we have determined an optimal pre-treatment protocol complemented with specific RT-qPCR reagents, that generates results equivalent to standard methods that involve RNA extraction. Minimizing RNA loss through the low heat approach, appropriate dilution of inhibitory substances and the higher sensitivity of TaqPath™ 1-Step RT-qPCR Master Mix may have played a combinatory role in achieving equivalency of the direct RT-qPCR compared to a standard approach requiring viral RNA extraction. By the direct approach, the greater loss of RNaseP $C_T$ compared to SARS-CoV-2 $C_T$ may be due to the fact that heating of specimens at 65˚C is more efficient for lysing viruses than nasopharyngeal epithelial cells and that a significant fraction of viruses remains extracellularly. Indeed, we have observed that SARS-CoV-2 is distributed approximately 50:50 between the supernatant and pellet after centrifugation of NPFS specimens (data not shown).

In summary, our new approach demonstrated high sensitivity and specificity in the detection of SARS-CoV-2 RNA, and the rate of RT-qPCR inhibition was similar to that of a standard approach. By skipping the RNA extraction step, the new approach will also significantly reduce the cost and improve the turn-around time of the assay. Delayed and inadequate laboratory testing can significantly hamper efforts to control the pandemic. Our results will help many labs all over the world who are struggling with a shortage of reagents to continue testing for SARS-CoV-2. Also, because of a significant reduction in cost, the optimized direct approach we described, will be useful to resource-limited countries to expand their capacity for RT-qPCR testing.

## Supporting information

**S1 Table. Primers and probes used in this study.**
(DOCX)

**S2 Table. Direct RT-qPCR on a nasopharyngeal specimen positive for human coronavirus HKU1 after heat treatment at 100˚C for 5 minutes.** NPFS specimens were either i) subjected to viral RNA extraction by standard method using a NucliSENS easyMAG automated extraction system (Biomerieux), or ii) serially diluted with nuclease free water (NFW) followed by incubation at 100˚C for 5 minutes, or iii) freeze thawed once prior to heat treatment. All heat-treated samples were centrifuged at 13,000 rpm for 5 minutes at 4˚C and supernatants were collected. All samples were tested for SARS-CoV-2 RNA by standard RT-qPCR in duplicate and mean $C_T$ values were compared.
(DOCX)

**S3 Table. Direct RT-qPCR on simulated, dry NPFS using a spiked specimen positive for SARS-CoV-2.** Specimens were collected from laboratory members who volunteered to provide specimens. NPFS specimens were either i) collected in VTM or ii) collected in sterile empty tubes as dry swabs and later resuspended in 1 ml of NFW. Five μl of a patient specimen positive for SARS-CoV-2 RNA ($C_T$ = 22) were spiked into 0.2 ml of specimens collected in

VTM of NFW. VTM specimens were extracted by standard method using a NucliSENS easy-Mag automated extraction system (bioMerieux). Specimens collected in NFW were incubated at 100˚C for 5 minutes and centrifuged at 13,000 rpm for 5 minutes at 4˚C and supernatants were collected. All samples were tested for SARS-CoV-2 RNA by standard RT-qPCR using Quantifast Pathogen RT-PCR + IC Master Mix in duplicate and mean $C_T$ values were compared.
(DOCX)

**S4 Table. Direct RT-qPCR on SARS-CoV-2 positive and negative NPFS specimens processed using Arcis Pathogen Kit.** NPFS specimens were processed according to manufacturer's protocol (Arcis Biotechnology Ltd.). All samples were tested for SARS-CoV-2 RNA by standard RT-qPCR in duplicate and mean $C_T$ values were compared.
(DOCX)

**S5 Table. Direct RT-qPCR on SARS-CoV-2 positive and negative NPFS specimens processed using Takara PrimeDirect Probe RT-qPCR mix.** NPFS specimens were processed according to manufacturer's protocol (PrimeDirect™ Probe RT-qPCR Mix, Takara Bio Inc.). All samples were tested for SARS-CoV-2 RNA by standard RT-qPCR in duplicate and mean $C_T$ values were compared.
(DOCX)

**S6 Table. Direct RT-qPCR on SARS-CoV-2 positive and negative NPFS specimens with or without heating at 65˚C for 5 minutes or after lysis using a non-ionic detergent.** NPFS specimens were either i) subjected to viral RNA extraction by standard method using a NucliSENS easyMAG automated extraction system (bioMerieux), or ii) diluted 4-fold with nuclease free water (NFW) iii) diluted 4-fold with nuclease free water (NFW) followed by incubation at 65˚C for 5 minutes or iv) diluted 4-fold with Tween-20 to final concentration of 0.2% followed by incubation at room temperature for 10 min. All samples were tested for SARS-CoV-2 RNA by standard RT-qPCR using Quantifast Pathogen RT-PCR + IC Master Mix.
(DOCX)

**S7 Table. Direct RT-qPCR on QCMD EQA specimens.** QCMD EQA specimens were simultaneously tested by standard and direct approach and by QIAstat-Dx Respiratory 2019-nCoV Panel (Qiagen).
(DOCX)

**S8 Table. Detection of SARS-CoV-2 with RNaseP as endogenous control by direct RT-qPCR and standard methods in NPFS specimens.**
(DOCX)

## Acknowledgments

We are grateful for the efforts of all technologists in the Molecular Infectious Disease Laboratory at Sidra Medicine.

## Author Contributions

**Conceptualization:** Mohammad Rubayet Hasan, Patrick Tang.

**Formal analysis:** Mohammad Rubayet Hasan, Hadi Mohamad Yassine, Andres Perez-Lopez.

**Investigation:** Sathyavathi Sundararaju, Hadi Mohamad Yassine, Andres Perez-Lopez, Patrick Tang.

**Methodology:** Faheem Mirza, Hamad Al-Hail, Sathyavathi Sundararaju, Thabisile Xaba, Muhammad Iqbal, Hashim Alhussain, Andres Perez-Lopez.

**Resources:** Hadi Mohamad Yassine.

**Supervision:** Mohammad Rubayet Hasan.

**Validation:** Faheem Mirza, Hamad Al-Hail, Thabisile Xaba, Muhammad Iqbal.

**Writing – original draft:** Mohammad Rubayet Hasan.

**Writing – review & editing:** Mohammad Rubayet Hasan, Andres Perez-Lopez, Patrick Tang.

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
