## [Decision Letter · Decision Letter 0]

18 Jun 2020

PONE-D-20-13888

Detection of SARS-CoV-2 RNA by direct RT-qPCR on nasopharyngeal specimens without extraction of viral RNA

PLOS ONE

Dear Dr. Hasan

Thank you for submitting your manuscript to PLOS ONE. After careful consideration, we feel that it has merit but does not fully meet PLOS ONE’s publication criteria as it currently stands. Therefore, we invite you to submit a revised version of the manuscript that addresses the points raised during the review process.

#1 For accuracy purposes the RTqPCR Assays aimed at detecting SAR COV2 RNA should involve a bona fide internal RNA control and not an external control as described in the ms.

We look forward to receiving your revised manuscript.

Kind regards,

Jean-Luc EPH Darlix, MG, Ph.D.

Academic Editor

PLOS ONE

Journal Requirements:

2. In the ethics statement in the manuscript and in the online submission form, please provide additional information about the patient records used in your retrospective study, including: a) whether all data were fully anonymized before you accessed them; and b) the source of the patient samples analyzed in this work (e.g. hospital, institution or medical center name).

3. To comply with PLOS ONE submission guidelines, in your Methods section, please provide additional information regarding your statistical analyses. For more information on PLOS ONE's expectations for statistical reporting, please see https://journals.plos.org/plosone/s/submission-guidelines.#loc-statistical-reporting.

4. We suggest you thoroughly copyedit your manuscript for language usage, spelling, and grammar. If you do not know anyone who can help you do this, you may wish to consider employing a professional scientific editing service.  

5. Please note that in order to use the direct billing option the corresponding author must be affiliated with the chosen institute. Please either amend your manuscript or remove this option (via Edit Submission).

Reviewers' comments:

Reviewer's Responses to Questions

**Comments to the Author**

1. Is the manuscript technically sound, and do the data support the conclusions?

Reviewer #1: Yes

2. Has the statistical analysis been performed appropriately and rigorously? 

Reviewer #1: Yes

3. Have the authors made all data underlying the findings in their manuscript fully available?

Reviewer #1: Yes

4. Is the manuscript presented in an intelligible fashion and written in standard English?

Reviewer #1: Yes

5. Review Comments to the Author

Reviewer #1: PONE D-20-13888

Detection of SARS-CoV-2 RNA by direct RT-qPCR on nasopharyngeal Specimens without extraction of viral RNA

reviewer's remarks

The study from Hasan et al describes a technique to facilitate detection of SARS-CoV2 RNA from biological specimens. By exploring different heating procedures and different qPCR mix reagents, the authors propose an optimized protocol that allows a sensitive detection of SARS RNA without the need of viral extraction. The study is well written and statistically convincing. Due to the importance of the pandemy and the extreme need of tests at the time of reviewing, this technical message should be rapidly transmitted to scientists and medical centers worldwide.

This information may indeed be useful to accelerate COVID diagnosis, especially in laboratories lacking ressources or reagents.

Point 1

However, a point could be adressed to improve the manuscript and increase its relevance in the field.

There is no doubt that the protocol described by Hasan et al can indeed be a relevant alternative to RNA-extraction based procedures.

Nevertheless, most qPCR tests aiming at detecting COVID infection extract RNA from Nasopharyngeal swabs and next measure the quantity of SARS-Cov2 RNA in the specimen AND the quantity of a cellular RNA that is not linked to infection. Human P RNase transcript is commonly choosen in several commercial kit as a control: this additionnal assay ensures that the amplification process is optimal, and checks that the biological specimens contains indeed a detectable material.

The assays presented by Hasan et al included internal control assay based on MS2 bacteriophage (either a bacteriophage preparation or an extracted MS2 bacteriophage RNA preparation) which is useful to validate the amplification reaction. However, this artificial addition provides no information about the quality of the specimen. Should a specimen being identified as COVID-negative by Hasan et al, there is no evidence that this sample contains anything detectable and has been properly collected.

To fully complete their study, the authors could show that their heating procedure is also effective to allow detection of the control P-RNase RNA or any human transcript serving as a control to validate the quality of the tested sample.

Conceptually it is probable that the heating protocol described in the manuscript may degradate softly the biological samples, which release some cellular and viral RNAs. This action should then render all RNAs accessible to the detection reaction. In that case authors could be able to show it or include this information in the manuscript.

However, we can reasonnably speculate that the viral cores present in a given sample would be more fragile than cellular membranes, and that the heating process is mainly efficient to release viral RNAs, and poorly efficient to release cellular ones (controls). This question -if not adressed by experimental data- should at least be discussed.

Point2

Beyond SARS-COV2, sample heating is commonly used to release RNAs from samples containing virus (HPV, BMYV, BChV...). Authors could mention some of them in the reference section.

6. PLOS authors have the option to publish the peer review history of their article (what does this mean?). If published, this will include your full peer review and any attached files.

Reviewer #1: No

---

## [Author Response · Author response to Decision Letter 0]

22 Jun 2020

PLOS ONE submission, PONE-D-20-13888

Point by point responses to reviewer’s comments

We would like to thank the editor and reviewer for taking the time to review this work and to provide constructive suggestions to improve the quality of our manuscript. Below, please find point by point responses to the reviewers’ comments:

Academic Editor

#1 For accuracy purposes the RTqPCR Assays aimed at detecting SAR COV2 RNA should involve a bona fide internal RNA control and not an external control as described in the ms.

>>We thank the editor for this comment. We agree with both the editor and the reviewer that some commercial assays used RNaseP as a specimen quality control (as well as RT-qPCR inhibition control) in their multiplex SARS-CoV-2 PCR assays. However, to our knowledge, this is not universally applied as seen in some FDA-approved assays such as Xpert- SARS-CoV-2 tests (Cepheid) or QIAstat-Dx® Respiratory SARS-CoV- 2 Panel (Qiagen), which do not include RNaseP or similar targets as a mrker for specimen quality. Clinical Laboratory Standards Institute (CLSI) and College of American Pathologists (CAP) also do not have any specific requirement for specimen quality control or cell control for respiratory specimens. 

However, in order to address this issue, we performed additional testing with RNaseP as internal control (as a duplex PCR with SARS-CoV-2) and added new data and discussion on this in the revised manuscript (page 10; lines 216-223 and page 11; lines 237-242).

Please also see our response to Reviewer # 1

Reference: 

1. CLSI. Molecular Diagnostic Methods for Infectious Diseases. 3rd ed. CLSI report MM03. Wayne, PA: Clinical and Laboratory Standards Institute; 2015.

2. College of American Pathologists, Microbiology checklist: CAP accreditation program; 2020

Reviewer #1 

reviewer's remarks

The study from Hasan et al describes a technique to facilitate detection of SARS-CoV2 RNA from biological specimens. By exploring different heating procedures and different qPCR mix reagents, the authors propose an optimized protocol that allows a sensitive detection of SARS RNA without the need of viral extraction. The study is well written and statistically convincing. Due to the importance of the pandemy and the extreme need of tests at the time of reviewing, this technical message should be rapidly transmitted to scientists and medical centers worldwide.

This information may indeed be useful to accelerate COVID diagnosis, especially in laboratories lacking ressources or reagents.

>>We thank the reviewer for these encouraging comments.

Point 1

However, a point could be adressed to improve the manuscript and increase its relevance in the field.

There is no doubt that the protocol described by Hasan et al can indeed be a relevant alternative to RNA-extraction based procedures.

Nevertheless, most qPCR tests aiming at detecting COVID infection extract RNA from Nasopharyngeal swabs and next measure the quantity of SARS-Cov2 RNA in the specimen AND the quantity of a cellular RNA that is not linked to infection. Human P RNase transcript is commonly choosen in several commercial kit as a control: this additionnal assay ensures that the amplification process is optimal, and checks that the biological specimens contains indeed a detectable material.

The assays presented by Hasan et al included internal control assay based on MS2 bacteriophage (either a bacteriophage preparation or an extracted MS2 bacteriophage RNA preparation) which is useful to validate the amplification reaction. However, this artificial addition provides no information about the quality of the specimen. Should a specimen being identified as COVID-negative by Hasan et al, there is no evidence that this sample contains anything detectable and has been properly collected.

To fully complete their study, the authors could show that their heating procedure is also effective to allow detection of the control P-RNase RNA or any human transcript serving as a control to validate the quality of the tested sample.

Conceptually it is probable that the heating protocol described in the manuscript may degradate softly the biological samples, which release some cellular and viral RNAs. This action should then render all RNAs accessible to the detection reaction. In that case authors could be able to show it or include this information in the manuscript.

However, we can reasonnably speculate that the viral cores present in a given sample would be more fragile than cellular membranes, and that the heating process is mainly efficient to release viral RNAs, and poorly efficient to release cellular ones (controls). This question -if not adressed by experimental data- should at least be discussed.

>>We agree with the reviewer that RNaseP is used as a specimen quality control (as well as RT-qPCR inhibition control) in many commercial multiplex assays for SARS-CoV-2, although it is not universally practiced. As the reviewer suggested, we have performed additional experiments to detect RNaseP (as a duplex PCR with SARS-CoV-2) by the direct method compared to the standard method that involves RNA extraction (Table S8). The results are consistent with reviewer’s prediction. In 30 NPFS specimens, RNaseP was detected in all specimens but with an average loss of CT of 4.2�1.7 as compared to an average loss of SARS-CoV-2 CT of 2�1.3. Interestingly, RNaseP was undetectable in two samples by the standard method, which could potentially be because of interfering substances concentrated by the extraction process. One sample that was positive for SARS-CoV-2 by the standard method with high CT (36.3) was missed by the direct method. We have added these results and discussions in the revised manuscript (page 10; lines 216-223 and page 11; lines 237-242)

Point2

Beyond SARS-COV2, sample heating is commonly used to release RNAs from samples containing virus (HPV, BMYV, BChV...). Authors could mention some of them in the reference section.

 >>We thank the reviewer in pointing out these previous studies. We have now mentioned about some of these relevant studies in the revised manuscript (Page 3; lines 57-61).

---

## [Editor Report · Decision Letter 1]

10 Jul 2020

Detection of SARS-CoV-2 RNA by direct RT-qPCR on nasopharyngeal specimens without extraction of viral RNA

PONE-D-20-13888R1

Dear Dr. Hasan

We’re pleased to inform you that your manuscript has been judged scientifically suitable for publication and will be formally accepted for publication once it meets all outstanding technical requirements.

Kind regards,

Jean-Luc EPH Darlix, MG, Ph.D.

Academic Editor

PLOS ONE
---

## [Editor Report · Acceptance letter]

17 Jul 2020

PONE-D-20-13888R1 

Detection of SARS-CoV-2 RNA by direct RT-qPCR on nasopharyngeal specimens without extraction of viral RNA 

Dear Dr. Hasan:

I'm pleased to inform you that your manuscript has been deemed suitable for publication in PLOS ONE. Congratulations! Your manuscript is now with our production department. 

Kind regards, 

on behalf of

Professor Jean-Luc EPH Darlix 

Academic Editor

PLOS ONE